# Manifold Distance Judge, an Adversarial Samples Defense Strategy Based on Service Orchestration

## Abstract

Deep neural networks (DNNs) are playing an increasingly significant role in the modern world. However, they are weak to adversarial examples that are generated by adding specially crafted perturbations. Most defenses against adversarial examples focused on refining the DNN models, which often sacrifice the performance and computational cost of models on benign samples. In this paper, we propose a manifold distance detection method to distinguish between legitimate samples and adversarial samples by measuring the different distances on the manifold. The manifold distance detection method neither modifies the protected models nor requires knowledge of the process for generating adversarial samples. Inspired by the effectiveness of the manifold distance detection, we demonstrated a well-designed orchestrated defense strategy, named Manifold Distance Judge (MDJ), which selects the best image processing method that will effectively expand the manifold distance between legitimate and adversarial samples, and thus, enhances the performance of the following manifold distance detection method. Tests on the ImageNet dataset, the MDJ is effective against the most adversarial samples under whitebox, graybox, and blackbox attack scenarios. We show empirically that the orchestration strategy MDJ is significantly better than Feature Squeezing on the recall rate. Meanwhile, MDJ achieves high detection rates against CW attack and DI-FGSM attack.

## 1 Introduction

Deep Neural Networks (DNNs) is the main research direction in artificial intelligence in recent years. Research and applications have shown that deep learning has replaced previous related technologies, and extraordinary breakthroughs have been made in image recognition, security-sensitive applications, and other fields(Kaiming He & Sun, 2016). However, DNNs are weak to adversarial attacks(Christian Szegedy & Fergus, 2014). Attackers can deceive the DNNs model by manipulating graph structure or node features, which results in graph counter disturbances, and limits their application in safety-critical systems. We can divide the attack into two different attack methods based on the attack principles. One is the gradient attack based on the known optimization model, and another one is based on the transfer attack model.

Recent researches have shown that we can use the Manifold Hypothesis theory to explain deep learning models from the mapping of data in high-dimensional space and low-dimensional space(Fefferman et al., 2016)The Manifold Hypothesis states that real-world high-dimensional data lie on low-dimensional manifolds embedded within the high-dimensional space(Aamari, 2019). This Hypothesis is important in DNN. Manifold hypothesis can explain the learning and detection process of deep learning models in image classification, as well as the principle of attacking against attacks(Fefferman & Narayanan, 2019). The Manifold Hypothesis explains why machine learning techniques are able to find useful features and produce accurate predictions from datasets that have a potentially large number of dimensions. From another perspective, the manifold hypothesis also explains why the adversarial attack can attack successfully and achieve the effect of confusing the deep learning model.

We can use the theory of manifold hypothesis to explain the three adversarial defense methods. Current defenses against adversarial examples follow three approaches: (1) Training the target classifier with adversarial examples, called adversarial training(Christian Szegedy & Fergus, 2014; Ian J. Goodfellow & Szegedy, 2015); (2) Making target classifiers hard to attack by blocking gradient pathway(Papernot et al., 2016b), e.g., defensive distillation and (3) training a classifier to distinguish between normal and adversarial examples(Fischer et al., 2017), e.g., Preprocessing.

In the analysis of defense methods using the manifold hypothesis, we can prove that in the deep learning model, the accuracy and generalization of the model cannot be taken into account at the same time. That is, when we improve the performance of one aspect, we will sacrifice another one. It also illustrates the necessity of building an adversarial sample defense system. Finally, we can also conclude that it is inevitable to produce adversarial samples. On the other hand, in systematic adversarial sample detection methods, there are also methods such as adversarial training, defensive distillation and preprocessing method. Among these methods, there are optimal defense methods for different attacks.Based on the previous manifold analysis and defense methods analysis, we propose an orchestration defense strategy, Manifold Distance Judge (MDJ), which is combining the manifold and the image preprocessing method.

This paper makes four contributions. First, through the analysis of the manifold hypothesis, we propose the manifold distance detection method to reduce the pass rate of adversarial samples by expanding the manifold distance between legitimate and adversarial samples. Second, simple image transformation can enhance the difference in feature distribution between adversarial samples and legitimate samples. Third, we propose a well-designed orchestration strategy, MDJ, for detecting adversarial examples. MDJ combines the manifold distance detection method and input image preprocessing method to obtain a high detection rate of adversarial samples. Meanwhile, the MDJ method we propose can be used as a supplement to other defense methods. Because the original model does not have to be changed, it is easy to combine with other defenses such as a detection method specifically for a certain type of attack. Moreover, we prove that there is an orchestration strategy which is a simple defense combination method in the image field, and its effect is better than the single simple defense, and the random combination of the weak defense combination method.

## 2 RELATED WORK

In this section, we classify adversarial attack methods and briefly summarize the defense methods of adversarial training, defensive distillation, and preprocessing-based techniques.

### 2.1 ADVERSARIAL SAMPLE GENERATION ALGORITHM

***Gradient Optimization Model:*** In a gradient optimization attack model, the attacker knows all the information and parameters inside the model and can generate adversarial samples based on the gradient of a given model to attack the classification model(Muñoz-González et al., 2017). We discuss the attacking algorithms used in our experiments, Fast Gradient Sign Method (FGSM)(Goodfellow et al., 2014), Basic Iterative Method (BIM)(Kurakin et al., 2018), DeepFool(Moosavi-Dezfooli et al., 2016), Acobian Saliency Map Approach (JSMA)(Papernot et al., 2016a), Carlini/Wagner Attacks(Carlini & Wagner, 2017).

***Transfer Attacks:*** In the transfer attack model, because the attacker does not know the internal information of the model, it is necessary to query the target model multiple times, use the alternative model and other optimization methods to solve the parameters based on the query results, generate the adversarial sample according to the gradient of the alternative model, and classify model attacks. we discuss the attacking algorithms used in our experiment, Momentum Iterative Fast Gradient Sign Method (MI-FGSM)(Dong et al., 2018),Diverse Input Iterative Fast Gradient Sign Method (DI$^2$-FGSM)(Xie et al., 2019).

### 2.2 EXISTING DETECTION METHODS

***Adversarial Training:*** The defense method is to superimpose the disturbance that maximizes the loss function in the sample, so it is effective for fixed or large disturbances, even if the attack method with obvious changes in the model gradient is easy to be detected.But adversarial training diminishes

the ML model's accuracy and can make the ML model more exposed to generalization(Carmon et al., 2019). Another disadvantage of Adversarial training based defense techniques is that we need to retrain the model whenever some new attack samples are discovered. It will be hard to update all deployed ML models.

***Defensive Distillation:*** The essence of the defense method is to replace the model defense idea, that is, to train a new distillation network based on the original model and its output to classify or predict samples.But distillation techniques work by combining the double model, and the second model uses the first model knowledge to improve accuracy. The black-box attack's recent improvement makes this out-of-date defense (Chakraborty et al., 2018). The strong transfer-potential of adversarial samples across neural network models (Papernot et al., 2016c) is the main reason for this method's collapse. It is not robust as simplistic variation in a neural network can make the system exposed to attacks (Carlini & Wagner, 2016). (He et al., 2017) concluded that combining/ensemble weak defenses does not automatically improve a system's robustness. Also, the ensemble technique remains static and vulnerable to a new attack.

***Processing-based Techniques:*** The defense method based on image preprocessing has a defensive effect on migration attacks and gradient attacks. Its essence is to reduce the noise of the sample, to reduce the interference of the adversarial disturbance to the model. For the choice of preprocessing method, we cannot judge which attack is effective, and most of the selection of preprocessing method adopts enumeration method, which is complicated to calculate.

## 3   MANIFOLD HYPOTHESIS AND ADVERSARIAL ATTACK

In this section, we introduce the manifold hypothesis to explain the learning and detection process of the deep learning model, and discuss the effect of this hypothesis in adversarial attacks.

### 3.1   MANIFOLD HYPOTHESIS

The Manifold Hypothesis states that real-world high-dimensional data lie on low-dimensional manifolds embedded within the high-dimensional space(Fefferman et al., 2016). It reflects the local smoothness of the decision function, helps to more accurately characterize the local area, and enables the decision function to better perform data simulation. It also explains why deep learning techniques are able to find useful features and produce accurate predictions from datasets that have a potentially large number of dimensions(Fefferman et al., 2016). The fact that the actual data set of interest actually lives on in a space of low dimension, means that a given deep learning model only needs to learn to focus on a few key features of the dataset to make decisions. Many of the algorithms behind machine learning techniques focus on ways to determine these (embedding) functions.

### 3.2   MANIFOLD HYPOTHESIS APPLIED IN ADVERSARIAL DEFENSE

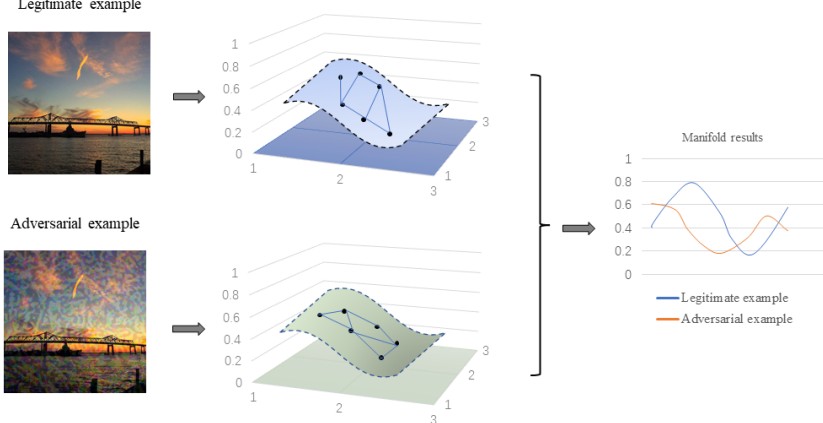

Figure 1: Manifold learning in defense

In the training stage, the models learn the manifold representation of the data and the optimal decision border of classification, taking classification task as an example. The model continuously maps different images on low dimensions to form a manifold and learns by confirming tags. In the model checking stage, the features of the picture are also mapped on low dimensions, and the distance from the manifold determined in the learning process is measured to determine the label output by the model. In the detection stage, the model extracts the characteristics of the input picture and compares it with the stored optimal decision border to determine the classification of the input picture. The optimal decision border affects the classification efficiency of the model. The reason is that too few data points cause the boundary shape to change during smoothing, which makes the appearance of adversarial samples inevitable; on the other hand, due to the excessive pursuit of the accuracy of the boundary, it leads to overfitting, and reduce the generalization and robustness of the model.

From the perspective of binary classification, there is no difference between adversarial samples and correctly classified negative samples. The reason is legitimate samples are extremely sparse in the distribution of high-dimensional space in the real world (take images as an example, MxNx8 black and white images, only the value in the high-dimensional extremely sparse field is meaningful). So we can use low-dimensional manifolds to classify legitimate samples. For deep learning models, if the adversarial samples and legitimate samples usually do not appear in which the samples are discriminated in high-dimensional space, it can be found that they are closer to the manifold than the legitimate negative samples. It leads to the failure to correctly classify legitimate samples into negative samples on the low-dimensional manifold that successfully classifies them.

From the perspective of two attack optimization models, there are two reasons for a classifier misclassifies an adversarial example: (1) The adversarial example is far from the boundary of the manifold of the task. (2) The adversarial example is close to the boundary of the manifold. If the classifier generalizes poorly off the manifold in the vicinity of the adversarial example, then misclassification occurs. The distribution probability of the adversarial sample and the legitimate sample is different, that is, the model decision probability of the adversarial sample and the legitimate sample should be different.

There is a theoretical method that can distinguish between legitimate samples and adversarial samples, that is, using adversarial samples that do not appear in the high-dimensional space of legitimate samples for judgment. This type of attack method against samples is to cross the boundary of the manifold with the smallest distance, and their mapping positions in the low-dimensional space are different from the legitimate samples. We assume that if there is a method that can change the mapping position of the image in the high-dimensional space, and also change the mapping position of the adversarial sample in the low-dimensional space, we can let the model learn the manifold of these samples to compare them. Adversarial samples are distinguished and used as the basis for classifying and detecting samples.

### 3.3    BASIC IDEA OF ADVERSARIAL SAMPLE DETECTION

Regarding how the model classifies samples, the following propositions can prove that there is a detection method that can detect adversarial samples and explain the existing detection methods. We can assume that the manifold is proposition A: then there is manifold M and algorithm A, if x is distributed as P(X), then x is on M. The defense detection uses its negative proposition B: that is, there are manifold M* and algorithm A*. If the distribution of x is not P(X), then x is not on M*.

The classification model is easy to be attacked because the dimensionality of the manifold has been excessively reduced. Increasing the dimensionality of the manifold can distinguish adversarial samples, but it is difficult to solve the direction of increasing dimensionality. Based on the above analysis, we can see that it is feasible to use the manifold analysis deep learning model, and it is also feasible to apply the concept of the manifold to classification detection and to explain the distinction between legitimate samples and adversarial samples.

The focus of using the principle of manifold hypothesis and improving detection performance is how to make low-dimensional manifolds better express high-dimensional data. For each defense method, there should be a corresponding optimal defense attack method, and for each attack, there should be an optimal defense method. We take the analysis of three more common defense methods as examples from the perspective of manifold.

***Adversarial training:*** this method can only defend against attacks added in the training phase, the reason is the model extracts the features of the adversarial samples formed by the attack method during training and thus can perform defense.

***defensive distillation:*** whether it can build its optimized network is critical to the effectiveness of this defense method. This is determined by whether the optimized network can accurately obtain the optimal discrimination boundary. If the learning effect of the boundary is perfect, the performance of defense against known attacks will be greatly improved. But for unknown attacks, the sample classification effect still cannot be achieved.

***preprocessing methods:*** preprocessing technologies can defend against different attacks. This is because different preprocessing methods have different effects on the samples, leading to changes in the representation of the picture in the high-dimensional space. For different attack methods, the changes of the attack samples are not the same, and the characteristics of the samples in the low-dimensional space also change, resulting in different attack methods for defense.

Therefore, this paper uses the Heuristics method to learn the original high-dimensional sample space which changes in the low-dimensional manifold. We propose a new idea of service orchestration strategy to combine different methods to expand the manifold distance.

## 4 THE MEANING OF ORCHESTRATION STRATEGY

Regarding the attack method, whether it is an attack based on an optimization model or a migration model, the ultimate goal is to make the label classification of the attack sample different from the original correct classification. From the perspective of manifold analysis, the purpose of the attack method is to find the fastest direction of gradient descent, derive the gradient, so that the shape of the adversarial sample in the high-dimensional space changes. It can also affect the mapping graph of the sample in the low-dimensional space, so that the adversarial sample crosses the boundary of the graph classification, to achieve the goal of a successful attack.

The general idea of the orchestration strategy should be able to change the mapping position of the picture in the high-dimensional space through picture processing and display this change. Now that the defense method can be connected with the attack method through the manifold, how to use this idea becomes the key to the orchestration strategy. Since the probability output of the deep learning model can reflect the distance of manifold distance between different samples, combined with the above analysis of the preprocessing method that has the characteristics of defending against gradient optimization models and transfer attack models, it becomes an ideal orchestration strategy idea. Here, we propose the orchestration strategy Manifold Distance Judge (MDJ) and it uses the output of deep learning and preprocessing methods to expand the distance between manifolds. The MDJ was expressed in the form of combination methods in the experimentand achieved excellent results in the ImageNet dataset(Deng et al., 2009).

## 5 OUR METHODS: MANIFOLD DISTANCE JUDGE

In this section, we introduce the Manifold Distance Judge (MDJ) strategy. It uses the manifold detection method to learn the manifold of samples and uses image preprocessing methods to expand the diversity of manifolds between samples. Manifold Distance Judge is essentially an orchestration defense strategythat achieves the optimal defense effect according to the characteristics of different defense methods.

### 5.1 BASIC IDEA OF OUR METHODS

Manifold distance judge (MDJ) is a detector that can detect whether the input sample is adversarial. The detector uses the combination of manifold distance detection method and the image preprocessing changes to represent the subtle changes of the input samples in the high-dimensional space in the low-dimensional space, achieving the purpose of identifying adversarial samples.

The detector is inspired by manifold hypothesis orchestration strategy and can exist independently as a detection system. This allows our method to be combined with other defenses, promoted, and enhance detection performance. In the selection of detection methods, the focus of improving

performance lies in how to change the input mapping position in the high-dimensional space in deep learning and how to represent the change of position distance.

## 5.2 Detector Based on Preprocessing

To solve the problem of how to change the mapping position of the picture in the high-dimensional space, we get inspiration from the preprocessing process of the image. The transformation of the image can intuitively change the feature representation of the sample, that is, change the mapping position of the image from a high-dimensional space.

For legitimate samples, this transformation does not change the mapping position much. But for adversarial samples, different transformations change the spatial mapping of the picture a lot, which is verified in the subsequent experimental process. We list the preprocessing methods used as following:

***Random Zoom:*** (Guo et al., 2017) considered cropping and rescaling of an image as one of their defenses, which is effectively zooming in on a portion of the image, an approach that was defeated by (Athalye et al., 2018b). We reuse this as one of our defenses, where the distance from each edge of the image is cropped by U[10, 50], independently for each edge.

***Swirl:*** We introduce a simple defense which is to apply a weak swirl to the image, rotating the pixels around a randomly selected point in the image. The radius of intensity is randomly selected from U[10, 200], and strength from U[0.1, 2.0]. The angle of rotation, center of rotation, and radius of effect are all randomized(Raff et al., 2019).

***Median Filter:*** We use a simple median filter, and follow the same approach as the Guassian blur. The radius of the blur kernel is chosen from r $\leftarrow$ U[2, 5], with a 50% chance all channels are forced to use the same radius(Raff et al., 2019).

***JPEG Noise:*** Using lossy JPEG compression to introduce artifacts was introduced by (Goodfellow et al., 2016). Their work looked at how different values of the JPEG compression level (a range from 1 to 100) reduced the impact of adversarial attacks for different values of $\epsilon \leq 16$. However, it was subsequently defeated, having 0% effectiveness (Athalye et al., 2018a). When using this approach, we randomize it by selecting the compression level from U[55, 95].

### 5.2.1 Detector Based on Probability Divergence

To indicate the change of position distancewe propose a manifold distance detection method, which utilize the output probability of softmax layer of DNN model. The probability can measure the distance of the sample to the decision border. Most neural network classifiers implement the softmax function at the last layer.

$$\text{softmax}(l)_i = \frac{\exp(l_i)}{\sum_{j=1}^{n} \exp(l_i)}$$

The output of softmax is a probability mass function over the classes. The input to softmax is a vector L called logit. Let rank(L,i)be the index of the element that is ranked the $i^{th}$ largest among all the elements in L. given a normal example whose logit is L, the goal of the attacker is to perturb the example to get a new logit $L_2$ such that rank(L,1) $\neq$ rank($L_2$,1)

Use f(x) to represent the output of the last layer (softmax) of the neural network and x as the input of the neural network. For the categories that have been determined, the neural network has a corresponding output probability for each category. We believe that these probabilities represent the comparison between the model's features for the input x and the model's learned classification features. The size of the probability output represents the shape of the input x in the low-dimensional manifold mapping and the distance from this type of feature in the low-dimensional manifold mapping.

We use a simple method,

$$dx = p(max) - p(secondmax),$$

to reflect the transformation of the distance of the picture manifold. Using only a single probability means that the effect is not significant for adversarial attacks, especially for adversarial samples that

have greatly crossed the boundary, their spatial mapping is almost the same as that of legitimate samples. It is the basic idea of the manifold distance detection method.

# 6 EXPERIMENT

In this section, we explain the experiment results for single method and the combination method after orchestration, and the comparison with Feature Squeezing.

## 6.1 RESULT-PERFORMANCE OF SINGLE METHOD

Table 1: Single defense method in transfer attacks

| Attacks | | Math | JPEG Noise | Random Zoom | Swirl | Median Filter |
|---|---|---|---|---|---|---|
| Black box | MI-FGSM | 0.91 | 0.33 | 0.61 | 0.21 | 0.76 |
| | DI-FGSM | 0.86 | 0.5 | 0.68 | 0.41 | 0.77 |
| Grey box | MI-FGSM | 0.74 | 0.67 | 0.74 | 0.3 | 0.74 |
| | DI-FGSM | 0.65 | 0.44 | 0.47 | 0.67 | 0.86 |

Table 2: Single defense method in gradient optimization model attacks

| Attacks | | | Math | JPEG Noise | Random Zoom | Swirl |
|---|---|---|---|---|---|---|
| White box | $CWL_0$ | LL | 0.97 | 1.00 | 1.00 | 0.63 |
| | | Next | 0.35 | 0.89 | 0.95 | 0.37 |
| | $CWL_2$ | LL | 0.94 | 1.00 | 0.99 | 0.72 |
| | | Next | 0.76 | 0.91 | 0.96 | 0.71 |
| | $CWL_i$ | LL | 0.78 | 1.00 | 1.00 | 0.74 |
| | | Next | 0.24 | 0.94 | 0.97 | 0.46 |
| | FGSM | | 0.75 | 0.49 | 0.90 | 0.30 |
| | DeepFool | | 0.52 | 0.77 | 0.84 | 0.23 |
| | BIM | | 0.01 | 0.48 | 0.79 | 0.09 |

Table 1 and Table 2 shows the detection rates for successful adversarial examples for each attack method. It can be obtained from TABLE 1 that the mainfold detection method performs better under black-box and gray-box attacks, and the ability to reach a detection model above 0.65 thresholds is weak. In contrast, in the white-box scenario, the attack algorithm generated a powerful ability to counter the sample interference model, which also led to the failure of the mainfold detection method in some attack methods in TABLE2. For instance, the detection rate of BIM attacks is only 0.01. The reason is that BIM, as a variant of FGSM, has efficacious interference to the model, which makes it impossible to distinguish samples under the current threshold. Second, in a CW attack, the attack only changes a limited number of pixels. Moreover, in CW attack, it only changes a limited number of pixels, so the Next class as the targeted attack has a greater impact on the detection model, resulting in an increase in the detection rate of the least-likely class as the targeted attack. This is consistent with the observation result of the mainfold detection method in TABLE2, that is, the mainfold detection method can improve the accuracy of model detection to varying degrees.

The detection effect of the preprocessing method is the opposite of that of the math method, and it has a higher detection rate in the white-box scene. As the attack algorithm learns more about the model, the more the features of the images can be changed by the preprocessing method, which in turn affects the judgment of the model. The observations in TABLE1 and TABLE2 confirm our conclusion. Secondly, for different sample preprocessing methods, in white-box attacks, the detection rates of JPEG Noise and Random Zoom are significantly higher than those of Swirl. The reason is that these two processing methods can blur or enlarge the image as an integrated, and change the features more than the Swirl method. However, the JPEG Noise method has no significant effect on FGSM and BIM attacks. In the gray-box scenario, the Swirl preprocessing method has a superior detection rate under MI-FGSM attacks. It illustrates that in different scenarios, each attack method has the most effective preprocessing way for it, and every preprocessing method can change the manifold characteristics of the image to a certain extent.

## 6.2 RESULT-PERFORMANCE OF THE COMBINATION METHOD

Table 3: Performance of the combination method in transfer attacks

| Attacks | | Black box | | Grey box | |
|---|---|---|---|---|---|
| | | MI-FGSM | DI-FGSM | MI-FGSM | DI-FGSM |
| JPEG-Noise | Comb 1 | 0.85 | 0.95 | 0.70 | 0.79 |
| | Comb 2 | 0.85 | 0.95 | 0.81 | 0.86 |
| | Comb 3 | 0.94 | 0.95 | 0.85 | 0.84 |
| Random Zoom | Comb 1 | 0.78 | 0.91 | 0.82 | 1.00 |
| | Comb 2 | 0.88 | 1.00 | 0.96 | 0.86 |
| | Comb 3 | 0.97 | 1.00 | 0.96 | 0.86 |
| Swirl | Comb 1 | 0.88 | 0.95 | 0.86 | 0.86 |
| | Comb 2 | 0.88 | 0.95 | 0.85 | 0.68 |
| | Comb 3 | 0.97 | 0.95 | 0.85 | 0.72 |
| Median Filter | Comb 1 | 0.91 | 0.86 | 0.74 | 0.70 |
| | Comb 2 | 0.94 | 1.00 | 0.93 | 1.00 |
| | Comb 3 | 1.00 | 1.00 | 0.93 | 1.00 |

Table 4: Performance of the combination method in gradient optimization model attacks

| Attacks | | White box | | | | | | | | |
|---|---|---|---|---|---|---|---|---|---|---|
| | | $CWL_0$ | | $CWL_2$ | | $CWL_i$ | | FGSM | DeepFool | BIM |
| | | LL | Next | LL | Next | LL | Next | | | |
| JPEG-Noise | Comb 1 | 0.66 | 0.66 | 0.45 | 0.47 | 0.47 | 0.53 | 0.75 | 0.61 | 0.48 |
| | Comb 2 | 1.00 | 0.94 | 1.00 | 1.00 | 1.00 | 0.97 | 0.77 | 0.93 | 0.71 |
| | Comb 3 | 1.00 | 0.97 | 1.00 | 1.00 | 1.00 | 0.99 | 0.78 | 0.95 | 0.71 |
| Random Zoom | Comb 1 | 0.57 | 0.55 | 0.45 | 0.55 | 0.46 | 0.46 | 0.58 | 0.51 | 0.57 |
| | Comb 2 | 1.00 | 0.98 | 0.99 | 0.96 | 1.00 | 0.99 | 0.90 | 0.94 | 0.91 |
| | Comb 3 | 1.00 | 1.00 | 1.00 | 1.00 | 1.00 | 0.99 | 0.95 | 0.97 | 0.95 |
| Swirl | Comb 1 | 0.86 | 0.61 | 0.80 | 0.57 | 0.83 | 0.58 | 0.62 | 0.53 | 0.12 |
| | Comb 2 | 0.90 | 0.59 | 0.95 | 0.85 | 0.93 | 0.72 | 0.61 | 0.50 | 0.08 |
| | Comb 3 | 0.93 | 0.66 | 0.98 | 0.93 | 0.95 | 0.72 | 0.67 | 0.61 | 0.15 |

TABLE3 and TABLE4 show the detection result of using the orchestration detective strategy to successfully attack samples. Comb1, Comb2, and Comb3 respectively represent the three kinds of Manifold Distance Judge's orchestration strategies.

***Comb1*** Comb1 is only to set up the threshold after performing the same image input transformation listed on the top of the table and use manifold detection method to make judgments. ***Comb2*** Comb2 combines two single defense methods simply. In Comb2, we first performed image transformation on the samples and set up the threshold of manifold detection. After completing the image transformer, we checked the labels of the images to find out the examples which labels have changed before and after the transformation, as well as the examples whose output of softmax is higher than the threshold for filtering, to achieve the effect of classifying adversarial samples. ***Comb3*** Comb3 firstly sets up a threshold, and perform manifold detection method on input samples. We filter samples above the threshold, perform image transformation on samples below the threshold, judge the change of the sample label, and use manifold detection method to detect again.

For these three orchestration strategies, Comb1 is equivalent to a random combination method. Compared to Comb2, Comb3 not only has an extra level of detection process but also the two-level detection method uses the idea of orchestration.

The first column in TABLE3 and TABLE4 indicates different preprocessing methods. We can see that for each preprocessing method, the detection rates of Comb1, Comb2, and Comb3 are rising significantly. Meanwhile, in a white-box attack scenario, a reasonably orchestrated Comb3 defense strategy, such as the Random Zoom method used in the preprocessing stage, can achieve a detection rate of more than 95% for all attacks. In the black-box attack scenario, the Comb3 combined defense strategy uses methods, such as Random Zoom and Median Filter in the preprocessing stage, to achieve a detection rate of 85% for all attacks. It further confirms our conclusion that the Manifold Distance Judge method designed by applying orchestration can effectively improve the accuracy of the model.

In summary, Manifold Distance Judge is meaningful, because after the preprocessing transformation, the distance between the legitimate sample and the adversarial sample on the manifold changes differently. In the orchestration defense strategy, we compared the performance of Comb1, Comb2, and Comb3, and found that an orchestration strategy that carefully detects changes in feature distances, such as Comb3, can obtain the best detection effect.

### 6.3 THE COMPARISON

We compare our results with Feature Squeezing(Xu et al., 2017) in Table 5. Comb represent the Com3 in the previous Section4. We configured the Feature Squeezing detectors on ImageNet following the same experiment set-up d in Feature Squeezing paper and reported the detection performance with our target models and the detection dataset. It shows that our recall value on the same white box attack is significantly higher than feature squeezing, especially on CW attacks, we can detect all pairs of samples, and the best classification rate of feature squeezing is only 0.87.

The experimental effect proves that our orchestration defense strategy is better than other combined defense methods, which further verifies the contribution point 3.

Table 5: Contrast with Feature Squeezing

| Attacks | | Recall | |
|---|---|---|---|
| | | Comb | Feature Squeezing |
| $CWL_0$ | LL | 1.00 | 0.85 |
| | Next | 1.00 | 0.85 |
| $CWL_2$ | LL | 1.00 | 0.86 |
| | Next | 1.00 | 0.87 |
| $CWL_i$ | LL | 1.00 | 0.82 |
| | Next | 0.99 | 0.77 |
| FGSM | | 0.95 | 0.33 |
| DeepFool | | 0.97 | 0.72 |
| BIM | | 0.95 | 0.41 |

## 7 CONCLUSION

From TABLE1 to TABLE4, we can evaluate the classification efficiency of the orchestrated combined defense model. The experimental results illustrated that the performance of a single detection method was inferior or equivalent to that of an ensemble combination of detection methods. However, the performance of the orchestration detection method was far better than that, which shows that the defense performance of this method has been improved significantly. Meanwhile, there were different combinations in the orchestration detection methods. The different input transformation methods will affect the final detection effect, and the choice of math method will also affect the final detection effect. From their perspective, we optimized the arrangement. The input transformation chose the Median Filter, and the math method selected the difference method instead of the maximum value, which highlights the significance of the orchestration.

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

## A  EXPERIMENT

### A.1  EXPERIMENT SETUP

Threat model Depending on the attacker's knowledge of target model, there are three scenarios:

***Black-box attack:***  Based on the alternative model, the attacker does not know the parameters of target model,.

***White-box attack:***  Based on the gradient optimization model, the attacker knows the parameter of target model.

***Gray-box attack:***  Based on the gradient optimization model, except for the parameters, the attacker knows everything else about the target model.

Table 6: Success rate for each attack on ImageNet

| Configuration | Attack | $L_i$ | | $L_2$ | | $L_0$ | | FGSM | BIM | DeepFool |
|---|---|---|---|---|---|---|---|---|---|---|
| | Mode | Next | LL | Next | LL | Next | LL | | | |
| Success Rate | | 0.99 | 0.99 | 0.90 | 0.97 | 1.00 | 1.00 | 0.99 | 1.00 | 0.89 |

We evaluate the manifold distance judge on the attacks described and summarized in Table 6. For each targeted attack, we try two different targets: the Next class (t = L + 1 mod #classes), and the least-likely class (LL), t = min $(\widehat{y})$. Here t is the target class, L is the index of the ground-truth class and $\widehat{y}$ is the prediction vector of an input image. This gives eleven total attacks: the four untargeted attacks (FGSM, BIM, MI-FGSM and DI-FGSM), and two versions each of the three targeted attacks

(JSMA, $CW_\infty$, $CW_2$, and $CW_0$). We use the implementations of FGSM, and BIM provided by the Cleverhans library. For MI-FGSM, DI-FGSM and the three CW attacks, we use the implementations from the original authors.

We chose different kinds of preprocessing methods and regarded the output of the Softmax layer of the model as a manifestation of the manifold distance. In the selection of preprocessing methods, we have selected several preprocessing methods that are completely different for image processing and compared them to verify the effectiveness of our method to the greatest extent.

Dataset and Target model We use the ImageNet dataset for the image classification task, and we set up a pre-trained model with the state-of-the-art performance. Meanwhile, we use the output of the last logits layer of the model as the input of softmax to obtain the required probability to represent the distance between manifolds.

In the ImageNet dataset, the target model of white-box attacks is set to the MobileNets model, because MobileNets are widely used on mobile phones and their small and efficient design makes it easier to conduct experiments. For black box and gray box attacks, the InceptionV3 model was chosen because it has excellent classification accuracy and is widely used in the field of deep learning image classification.

Detection model We choose InceptionV4 as the gray-box attack detection model, and the ResNet152 model as the black-box because gray-box attacks between similar structural models are achievable. Another reason is the InceptionV4 and the InceptionV3 model have similar model structures, while the ResNet152 model is different. This model selection method can simulate gray box and black box attacks to the greatest extent.

Training As we mentioned before, in the image detection of the deep learning model, there are differences between the features of the normal image and the features of the adversarial sample. This is reflected in the probability output of the Softmax layer of the model, and the subsequent experimental part is represented by mathematic. Based on this feature, before each detection model starts to detect, 100 images that can be correctly identified are randomly selected in ImageNet, and they are input into the detection model to select the appropriate anti-sample filter index to select the threshold.

Validation Next, we use the chosen threshold value to measure the detection rate on three groups: successful adversarial examples (SAEs), failed adversarial examples (FAEs), and legitimate examples (for false positive rate). Except when noted explicitly, "detection rate" means the detection rate on successful adversarial examples. It is important to distinguish failed adversarial examples from legitimate examples here since detecting failed adversarial examples is useful for detecting attacks early, whereas an alarm on a legitimate example is always undesirable and is counted as a false positive.

## A.2 THE COMPARISON

Table 7 shows the detection effects of our different orchestration strategies on successful attack samples under the attack based on gradient optimization. Single represents the single method processing effect of the preprocessing method on the far right of the table. In Table 7, single represents a single method, and ensemble and comb represent an orchestration strategy. It can be seen from the difference between the recall and IS indicators that the effect of the orchestration method is significant due to the single method. Compared with a single defense method, in this experimental result, the recall and IS values of combination methods increase due to the change of the sample label of the attacking sample after the image transformation, indicating that more adversarial samples are detected.

Different orchestration strategies will produce different effects. On the one hand, for different detection methods, the effect of detecting the change of the characteristic distance of the sample is different. On the other hand, from the significance of the arrangement, whether the latter detection method can effectively use the conclusion of the previous level, that is, according to the pretreatment method of the previous stage to select the method of the latter stage.

Table 7: Performance of different orchestration strategies in gradient optimization model Attacks

| Transformers | | | acc | | | recall | | | IS | | |
|---|---|---|---|---|---|---|---|---|---|---|---|
| | | | Single | Comb2 | Comb3 | Single | Comb2 | Comb3 | Single | Comb2 | Comb3 |
| Random Zoom | $CWL_0$ | LL | 0.93 | 0.86 | 0.81 | 1.00 | 1.00 | 1.00 | 1.00 | 1.00 | 1.00 |
| | | Next | 0.94 | 0.86 | 0.81 | 0.95 | 0.98 | 1.00 | 0.90 | 0.95 | 1.00 |
| | $CWL_2$ | LL | 0.93 | 0.85 | 0.81 | 0.99 | 0.99 | 1.00 | 0.98 | 0.97 | 1.00 |
| | | Next | 0.92 | 0.82 | 0.80 | 0.96 | 0.96 | 1.00 | 0.91 | 0.89 | 1.00 |
| | $CWL_i$ | LL | 0.93 | 0.86 | 0.80 | 1.00 | 1.00 | 1.00 | 1.00 | 1.00 | 1.00 |
| | | Next | 0.94 | 0.86 | 0.83 | 0.97 | 0.99 | 0.99 | 0.94 | 0.97 | 0.97 |
| | FGSM | | 0.84 | 0.82 | 0.78 | 0.90 | 0.90 | 0.95 | 0.63 | 0.76 | 0.85 |
| | DeepFool | | 0.85 | 0.79 | 0.80 | 0.84 | 0.94 | 0.97 | 0.69 | 0.83 | 0.91 |
| | BIM | | 0.83 | 0.81 | 0.80 | 0.79 | 0.91 | 0.95 | 0.61 | 0.78 | 0.86 |
| Swirl | $CWL_0$ | LL | 0.80 | 0.80 | 0.80 | 0.63 | 0.90 | 0.93 | 0.44 | 0.75 | 0.81 |
| | | Next | 0.65 | 0.65 | 0.68 | 0.37 | 0.59 | 0.66 | 0.19 | 0.26 | 0.34 |
| | $CWL_2$ | LL | 0.83 | 0.82 | 0.83 | 0.72 | 0.95 | 0.98 | 0.55 | 0.86 | 0.94 |
| | | Next | 0.82 | 0.78 | 0.81 | 0.71 | 0.85 | 0.93 | 0.53 | 0.66 | 0.81 |
| | $CWL_i$ | LL | 0.83 | 0.81 | 0.82 | 0.74 | 0.93 | 0.95 | 0.56 | 0.81 | 0.86 |
| | | Next | 0.70 | 0.70 | 0.72 | 0.46 | 0.72 | 0.72 | 0.27 | 0.42 | 0.43 |
| | FGSM | | 0.63 | 0.65 | 0.69 | 0.30 | 0.61 | 0.67 | 0.15 | 0.27 | 0.37 |
| | DeepFool | | 0.60 | 0.60 | 0.64 | 0.23 | 0.50 | 0.61 | 0.11 | 0.17 | 0.27 |
| | BIM | | 0.54 | 0.40 | 0.43 | 0.09 | 0.08 | 0.15 | 0.04 | -0.12 | -0.10 |
| JPEG Noise | $CWL_0$ | LL | 0.95 | 0.83 | 0.82 | 1.00 | 1.00 | 1.00 | 1.00 | 1.00 | 1.00 |
| | | Next | 0.90 | 0.81 | 0.81 | 0.89 | 0.94 | 0.97 | 0.88 | 0.84 | 0.91 |
| | $CWL_2$ | LL | 0.96 | 0.85 | 0.83 | 1.00 | 1.00 | 1.00 | 1.00 | 1.00 | 1.00 |
| | | Next | 0.91 | 0.85 | 0.81 | 0.91 | 1.00 | 1.00 | 83.00 | 1.00 | 1.00 |
| | $CWL_i$ | LL | 0.95 | 0.84 | 0.82 | 1.00 | 1.00 | 1.00 | 1.00 | 1.00 | 1.00 |
| | | Next | 0.93 | 0.83 | 0.82 | 0.94 | 0.97 | 0.99 | 0.88 | 0.92 | 0.97 |
| | FGSM | | 0.70 | 0.71 | 0.72 | 0.49 | 0.77 | 0.78 | 0.27 | 0.48 | 0.50 |
| | DeepFool | | 0.84 | 0.83 | 0.79 | 0.77 | 0.93 | 0.95 | 0.60 | 0.82 | 0.85 |
| | BIM | | 0.69 | 0.70 | 0.67 | 0.48 | 0.71 | 0.71 | 0.26 | 0.40 | 0.37 |

