# OpenReview forum: "Manifold Distance Judge, an Adversarial Samples Defense Strategy Based on Service Orchestration"
_ICLR.cc/2022/Conference — ICLR 2022 Submitted_

### Official Review · Reviewer_MGG8 · 2021-10-23

**Correctness:** 3
**Technical Novelty And Significance:** 2
**Empirical Novelty And Significance:** 2
**Recommendation:** 5
**Confidence:** 4

**Main Review:**

The proposed method is based on the famous manifold hypothesis, that is the real-world high dimensional data generally lie in the lower dimensional manifold.

1: I think the discussion in Section 3 is trivial and well-known in the deep learning community, hence it is unnecessary to spend many contexts to discuss this.

2: May I ask about those experiments, how are the threat models defined? Against each adversarial attack, how is the attack defined? Could the authors provide some details on this?

**Summary Of The Paper:**

This paper studies the manifold distance detection between adversarial examples and clean data. The proposed Manifold Distance Judge has shown effectiveness in detecting adversarial examples under various adversarial attack settings.

**Summary Of The Review:**

Overall, I do not think the paper provides a significant contribution to the community. The proposed method is trivial and similar kinds of pre-processing methods have already been considered.

---

### Official Review · Reviewer_wmbn · 2021-11-03

**Correctness:** 2
**Technical Novelty And Significance:** 1
**Empirical Novelty And Significance:** 2
**Recommendation:** 1
**Confidence:** 5

**Main Review:**

[Strength]
1.	At least the experiments were done with various attack & defense methods. However, as mentioned in the comments, a proper set of competitors should be other cutting-edge manifold defense methods and the authors should perform additional experiments against an adversary stronger than the vanilla implementation.

[Weakness]
1.	The paper contains severe writing issues such as grammatical errors, abuses of mathematical symbols, unclear sentences, etc.
2.	The paper needs more literature survey, especially about the existing defense methods using the manifold assumption.
3.	The paper does not have enough (either theoretical or experimental) progress to get accepted, compared to previous methods using the manifold assumption.

[Comments]
1.	First of all, use some spell/grammar checker (or ask someone else to proofread) to fix basic grammatical errors.
2.	Section 3 is very unclear in general.
First, I cannot understand the reason why the Section is needed at all. Manifold-based defense against adversarial example is not a new approach and reviewers know well about the manifold assumption in the adversarial machine learning setting. Section 3 does not introduce anything new more than those reviewers’ understanding, and the reasonings are too crude to be called an “analysis”.
Second, the writing is not cohesive enough. Each paragraph is saying some topic, however, the connections between the paragraphs are not very clear, making Section 3 more confusing. Even in a single paragraph, the logical reasonings between sentences are sometimes not provided at all.
Third, some of those contents are added for no reason. For example, Figure 1 exists for no reason whereas the figure is not referred at all in the paper. The propositions mentioned in Section 3.3 are vaguely written and not used at all. By having these unnecessary parts, the writing looks to be verbose and overstating.
3.	In Section 5, the defense method should be written with more formality. Based on the description given in the paper “dx = p(max)-p(secondmax)”, it is very unclear what each term means. Each probability (the authors did not even say that they are probabilities) must correspond to the output from a softmax layer, but which model provides such a softmax layer output, the target classifier, or is there another classifier prepared for it? How are the described transformations used to get the divergence value? What does the detector do with the divergence? (All of these details should be described in Section 5.) Section 6.2 mentions some thresholding strategies, how did the detector work in Section 6.1, though? When thresholding is used, what is the threshold value used and what is the rationale of the choice of the threshold value? There are so many missing details to understand the method.
4.	Section 7 looks to be a conclusion for experiments. This should be moved to Section 6 and Section 7 should be an overall conclusion of the paper.
5.	The suggested method is neither creative nor novel, compared to the existing methods utilizing the distance from manifolds. As pointed out, the defense based on the manifold assumption is not a new approach. [1][3][4][6](These papers are only a few representative examples. There are many other papers on this type of defense.) Moreover, the idea of using probability divergence is already proposed by previous work [1] and an effective attack for such detection already exists. [2] (Of course, this paper proposes another probability divergence, but there is no support that this method could be significantly better than the previous work.)
6.	The experiment should be done more extensively. It looks like that some transformations were brought from the Raff et al. paper [5] which tested the defense against the adversary as strong as possible. Specifically, Raff et al. considered potential improvements of existing attacks to attack their work then tested the defense performance against the improved attack. However, the paper only uses vanilla implementation in the Cleverhans library (or by the original authors). The authors should have shown that the proposed method is robust against a stronger adversary because adversaries who are aware of the method will not use a simple version of the attack. (At least, those adversaries will try using the attack suggested by Raff et al.)

[References]

[1] (Meng & Chen) MagNet: a Two-Pronged Defense Against Adversarial Examples

[2] (Carlini & Wagner) MagNet and “Efficient Defenses Against Adversarial Attacks” are Not Robust to Adversarial Examples

[3] (Samangouei et al.) Defense-GAN: Protecting Classifiers Against Adversarial Attacks Using Generative Models

[4] (Jiang et al.) To Trust or Not to Trust a Classifier

[5] (Raff et al.) Barrage of Random Transforms for Adversarially Robust Defense

[6] (Dubey et al.) Defense Against Adversarial Images using Web-Scale Nearest-Neighbor Search

**Summary Of The Paper:**

Based on the manifold assumption in the adversarial machine learning setting, this paper proposes a detection strategy using a set of different image transformations and the classifier’s output probability divergence. Some experiments were performed to show the detection performance against vanilla implementations of well-known attack methods.

**Summary Of The Review:**

I believe that the paper quality is far below the acceptance threshold. First, this paper does not contain any significant progress in the manifold-based defense method. Second, the average quality of paper writing is extremely poor to understand the paper properly.

---

### Official Review · Reviewer_fqVb · 2021-11-03

**Correctness:** 2
**Technical Novelty And Significance:** 1
**Empirical Novelty And Significance:** 1
**Recommendation:** 1
**Confidence:** 4

**Main Review:**

**Strengths**
- The paper reports high detection rates of adversarial examples generated using a variety of attacks in the black-, grey- and white-box settings.
**Weaknesses**
- The quality of writing and explanation in this paper is rather poor, to the point that it actively impedes the reader's ability to understand the methods being proposed and the experiments that are conducted. My summary of the paper is an educated guess based on what I could parse from the language, but I may have misunderstood due to issues with the writing and presentation. I encourage the authors to improve the writing and presentation, specifically by giving a summary of the detection method used in the form of a algorithm block in the text.
- From the description of the white-box attacks in the paper, it would appear that the proposed detection method has not been validated with adaptive attacks. An indication of this comes from the detection performance against the Basic Iterative Method in Table 2. I suspect that with a more careful evaluation, the proposed detection method would not be successful. The fact that detection methods are not effective at dealing with adaptive attacks has been known for some time (see *Adversarial Examples Are Not Easily Detected: Bypassing Ten Detection Methods* by Carlini and Wagner). This paper could be drastically improved with a detailed analysis of an adaptive attack against the proposed method.
- The paper only considers a single source of information (the so-called manifold) to determine if an example is adversarial or not. It does not consider the use of embedding layers within the network etc. as other possible sources of information for detection.
- Crucial experimental details such as the parametrization of the attacks used are missing. Further, in some cases such as for the FGSM attack, the detection rate is lower for grey-box attacks than for more powerful white-box attacks, which is incorrect and points to experimental error.

**Summary Of The Paper:**

This paper proposes using the difference between the highest and second-highest softmax outputs from a model to detect adversarial examples. It also considers the use of different preprocessing methods to modify the output probabilities such that the adversarial examples can be well-differentiated from benign ones.

**Summary Of The Review:**

This paper suffers from poor quality of writing, an unclear contribution to the already vast literature on the detection of adversarial examples and limited engagement with well-known recommendations from the research community on the evaluation of defenses/detection methods for adversarial examples.
++++++++++++++++++++++
No response so I retain my score.

---

### Official Review · Reviewer_Cbtd · 2021-11-09

**Correctness:** 2
**Technical Novelty And Significance:** 2
**Empirical Novelty And Significance:** Not applicable
**Recommendation:** 3
**Confidence:** 4

**Main Review:**

This paper proposes a manifold distance based detection based against adversarial samples. In addition, it also proposes manifold distance judge for the adversarial defense. The focus of the paper is to make low-dimensional manifolds better express high-dimensional data; the experimental results seems that the proposed method has good performance. However, I have the following concerns:
1.	The paper introduces too much intuitive idea, but it lacks technical steps;
2.	Technical contributions should be better analyzed. Otherwise, it is hardly to find the contributions of the paper;
3.	The basic idea seems interesting, but the manifold distance method has already be utilized in many extant works;
4.	Why the authors only select Random zoom, Swirl, Median filter, and JPEG noise as the preprocessing methods? It is difficult to find the relation between the proposed detection method with the preprocessing methods;
5.	In the experiment, comparing with Feature squeezing seems inadequate. In addition, the experimental results should be better organized;
6.	There are some typos and grammar errors, such as “Table 1 and Table 2 show”, etc.


**Summary Of The Paper:**

This paper proposes a manifold distance based detection based against adversarial samples. In addition, it also proposes manifold distance judge for the adversarial defense. The focus of the paper is to make low-dimensional manifolds better express high-dimensional data; the experimental results seems that the proposed method has good performance, but there are still lots of space to improve.

**Summary Of The Review:**

This paper proposes some novel idea to defend adversarial attacks, but the paper should be better organized to present the technical contributions.

---

### Decision · Program_Chairs · 2022-01-20

**Decision:**

Reject

**Comment:**

The paper proposes a manifold distance-based detection based against adversarial samples, i.e., using the difference between the highest and second-highest softmax outputs from a model to detect adversarial examples. All the reviewers gave negative scores. The main concerns lie in 1) poor quality of writing; 2) contributions of the paper are not clearly stated; and 3) limited engagement with well-known recommendations from the research community on the evaluation of defenses/detection methods for adversarial examples. No rebuttals are provided. Thus, I cannot recommend accepting the paper to ICLR.